# Echocardiographic Assessment of Patients with Pulmonary Tumor Thrombotic Microangiopathy First Diagnosed in the Emergency Department

**DOI:** 10.3390/diagnostics12020259

**Published:** 2022-01-20

**Authors:** Minjoo Kim, Hee Yoon, Min Yeong Kim, Ik Joon Jo, Soo Yeon Kang, Guntak Lee, Jong Eun Park, Taerim Kim, Se Uk Lee, Sung Yeon Hwang, Won Chul Cha, Tae Gun Shin

**Affiliations:** 1Department of Emergency Medicine, Samsung Medical Center, Sungkyunkwan University School of Medicine, Seoul 06351, Korea; minjuu.kim@samsung.com (M.K.); ikjoon.jo@samsung.com (I.J.J.); syrei3.kang@samsung.com (S.Y.K.); guntak.lee@samsung.com (G.L.); jongeun7.park@samsung.com (J.E.P.); taerimi.kim@samsung.com (T.K.); seuk.lee@samsung.com (S.U.L.); sygood.hwang@samsung.com (S.Y.H.); wc.cha@samsung.com (W.C.C.); taegun.shin@samsung.com (T.G.S.); 2Department of Radiology, Samsung Medical Center, Sungkyunkwan University School of Medicine, Seoul 06351, Korea; minyeong1.kim@samsung.com

**Keywords:** pulmonary tumor thrombotic microangiopathy: echocardiography, right heart failure, pulmonary hypertension, malignancy

## Abstract

Pulmonary tumor thrombotic microangiopathy (PTTM) is a fatal disease that obstructs pulmonary vessels, leading to pulmonary hypertension (PH) and right-sided heart failure causing rapid progressive dyspnea in patients with cancer. This retrospective chart review involved nine patients with PTTM who were first clinically diagnosed in a tertiary emergency department (ED) between January 2015 and June 2021. They underwent laboratory tests, chest radiography, chest computed tomography (CT), and echocardiography. All patients presented with severe and rapidly progressive dyspnea within a few days, a high oxygen demand. The right ventricle (RV): left ventricle ratio was >1 on chest CT, and no life-threatening pulmonary thromboembolism (PTE) was observed. Echocardiographic findings indicated that all patients had moderate-to-severe RV dilatation with a D-shaped LV. The median tricuspid regurgitation maximum velocity was 3.8 m/s, and the median RV systolic pressure was 63 mmHg, indicating severe PH. The median value of tricuspid annular plane systolic excursion was 15 mm, showing a decrease in RV systolic function, and McConnell’s sign was observed in five patients. Two patients immediately underwent chemotherapy and are currently alive. PTTM should be suspected and evaluated using echocardiography in patients with cancer presenting to the ED with acute dyspnea and RV failure without PTE.

## 1. Introduction

Pulmonary tumor thrombotic microangiopathy (PTTM) is a rare and fatal disease that obstructs pulmonary vessels, leading to pulmonary hypertension (PH) and right-sided heart failure [1] that causes rapid progressive dyspnea in patients with cancer. PTTM is caused by tumor cell microemboli, leading to progressive vascular occlusion with obstructive fibro-intimal remodeling in tiny pulmonary arteries, veins, and lymphatic systems [2]. A confirmed diagnosis of PTTM requires cytological examination of the aspirate from a wedge-shaped pulmonary artery catheter and lung biopsy [2,3,4]. However, when a patient with PTTM presents to the emergency department (ED) with shortness of breath, exacerbation advances quickly and often leads to death, making ante-mortem diagnosis and treatment challenging. As the number of patients with PTTM increases along with increase in the number of patients with cancer, it is essential to recognize the clinical characteristics and echocardiographic findings of PTTM to initiate appropriate therapy in the ED.

Point-of-care ultrasound (POCUS) has been widely used worldwide as a valuable tool for rapid bedside evaluation and treatment of patients with dyspnea [5,6]. Adding POCUS to the usual diagnostic approaches that rely on radiological and laboratory results has been shown to improve diagnostic accuracy and reduce diagnostic time through bedside evaluations of thoracic lesions and heart function [7,8]. However, although PH and right-sided heart failure are well-known features of the clinical course of PTTM, no studies have reported detailed echocardiographic findings concerning patients with PTTM [9]. In this study, we have investigated the clinical and echocardiographic characteristics of patients with PTTM first clinically diagnosed with PTTM in the ED. Based on specific echocardiographic findings, we recommend the use of echocardiography in conjunction with other tests to quickly and effectively diagnose PTTM in patients with cancer who present to the ED.

## 2. Materials and Methods

We conducted a retrospective chart review of nine patients with PTTM first clinically diagnosed in the ED of a tertiary academic medical center in South Korea from January 2015 to June 2021. We searched for patients with new-onset PH or thrombotic microangiopathy in the ED as there is no accurate diagnostic code for PTTM at our institution. We screened 60 patients, among whom 37 had no malignant disease and 12 had secondary PH related to other diseases. Excluding two patients for whom echocardiography data were not stored, we finally reviewed the data of nine patients relevant to our study ([Fig diagnostics-12-00259-g001]). This study was conducted in accordance with the Declaration of Helsinki and was approved by the Institutional Review Board (IRB) of Samsung Medical Center (IRB file number 2021-04-127).

### Data Collection

All patients had undergone several laboratory tests, electrocardiography, chest radiography, chest computed tomography (CT), and bedside echocardiography in the ED. Baseline demographic and clinical data on the following were collected: age, sex, presence of previously diagnosed PH and malignancy, time to exacerbation of dyspnea and oxygen demand, treatment history including chemotherapy trials after diagnosis of PTTM, and prognosis. Blood laboratory tests included complete blood count and electrolyte tests and basic chemistry tests. In particular, we reviewed data on D-dimer, cardiac enzyme, and NT-pro BNP levels as serologic markers related to the activation of coagulation and heart failure. On plain radiography, we checked for enlargement of the pulmonary trunk and lung abnormalities. On chest CT, the following features were evaluated: the right ventricle (RV): left ventricle (LV) ratio (RV: LV ratio), visible pulmonary thromboembolism (PTE), vascular tree-in-bud sign, centrilobular ground glass nodules (GGNs), peripheral ground-glass opacities (GGOs), interstitial thickening, and consolidation. To calculate the RV: LV ratio, the maximal diameters of both the right and left ventricular cavities were measured on axial CT scans (plain radiography and CT scans were reviewed by a radiology specialist, Dr. K.M.Y).

Echocardiography was performed by emergency physicians in the ED, and the stored images were reviewed by two POCUS specialist attending physicians (J.I.J and Y.H.). Echocardiographic parameters included chamber size and bilateral ventricular function. In particular, we focused on the presence or absence of RV systolic dysfunction by measuring tricuspid annular plane systolic excursion (TAPSE) using M-mode, lateral tricuspid annulus peak systolic velocity (s’) using tissue Doppler imaging, tricuspid regurgitation maximum velocity (TR Vmax) and right ventricular systolic pressure (RVSP) using color flow and spectral Doppler echocardiography. The presence of a D-shaped LV and McConnell’s sign was also assessed. LV systolic function was evaluated using a modified Simpson’s method or through visual estimation. Some patients had undergone an additional comprehensive echocardiography performed by a cardiologist. Echocardiography was done using a Samsung ultrasound HM70A with a 2–4 MHz cardiac transducer (Samsung Healthcare, Seoul, South Korea).

## 3. Results

### 3.1. Baseline Characteristics and Clinical Manifestations

The median age of the nine patients with PTTM was 53 years, and eight patients were female. Within a few days, all patients experienced severe and rapidly worsening dyspnea, and most had a high oxygen demand of >5 L/min via a facial mask (Appendix B). Among the seven patients with cancer, four had breast cancer, two had stomach cancer, and one had bladder cancer. Two patients were suspected of having cancer in the ED, but pathological confirmation was not possible due to a rapid deterioration in their clinical symptoms. After a clinical diagnosis of PTTM in the ED, two patients almost immediately underwent chemotherapy within 48 h, and they are currently alive and undergoing treatment. The remaining patients died within 3 days of hospital admission from the ED (Table 1).

### 3.2. Laboratory Tests, Plain Chest Radiography, and CT Findings

Cardiac enzyme levels concerning troponin T and D-dimer were both high in all nine patients. There were no specific findings other than right-axis deviation on the electrocardiogram. The median interquartile (IQR) D-dimer and N-terminal pro b-type natriuretic peptide (NT-proBNP) levels were particularly high at 16.52 (6.18–25.09) μg/mL and 4237 (3664–8153) pg/mL, respectively (Table 1). Plain chest radiography findings indicated that 50% of the patients had cardiomegaly and enlargement of the pulmonary trunk. Abnormalities of the lung, such as GGOs or interstitial thickening, were observed in only two patients, with most patients showing no new pulmonary lesions. On chest CT, specific findings of PTTM, such as a vascular tree-in-bud sign and centrilobular GGNs were identified in five and eight patients, respectively. In particular, all patients had an RV: LV ratio > 1, indicating RV dilation. Nonspecific findings included peripheral wedge GGOs, interstitial thickening, and consolidation. Except for one patient with a minor subsegmental PTE, all patients were grossly clear of PTE (Table 2, [Fig diagnostics-12-00259-g002]).

### 3.3. Echocardiography Findings

Echocardiography was performed by emergency physicians for all nine patients, and the characteristic findings are described in Table 3 and [Fig diagnostics-12-00259-g003]. All patients showed moderate-to-severe RV dilatation with D-shaped LV on echocardiography (Appendix A). The base and mid-RV median sizes (IQR) were 4.2 (4.2–4.5) cm and 3.4 (3.1–3.7) cm, respectively. The median TR Vmax (IQR) was 3.8 (3.5–3.8) m/s, and the median RVSP was 63 (58–79) mmHg, indicating severe PH. The median value of TAPSE (n = 7) was 15 (11–17) mm, showing a decrease in RV systolic function, with one patient having a normal value of 21 mm. The values of S’ (n = 4) ranged from 7.7 to 11.9 cm/s. Notably, five patients showed McConnell’s signs of RV dysfunction with a regional pattern of akinesia of the medial free wall, but normal motion at the apex (Appendix A). However, the LV size was normal in all patients, and the LV ejection fraction did not decrease significantly, ranging from 56% to 67%. The e/e′ values for LV diastolic dysfunction in four patients were within the normal range (5.7–8.9).

## 4. Discussion

This study involved a case series of patients with PTTM diagnosed for the first time in the ED and confirmed the characteristic echocardiographic findings of PH and right-sided heart failure. All patients showed moderate-to-severe RV dilatation with a D-shaped LV on echocardiography. Furthermore, there was a significant increase in pulmonary arterial pressure, and RV systolic dysfunction was confirmed without LV systolic dysfunction. However, none of the patients had visible PTE, which can cause hemodynamic instability. This case series showed that echocardiography is a valuable tool for evaluating heart function in patients with severe dyspnea [10], which may be performed immediately at the patient’s bedside and may be faster than other evaluation methods. Therefore, based on specific echocardiographic findings, we recommend the use of echocardiography in conjunction with other tests to diagnose PTTM quickly and effectively in patients with cancer who present to the ED with dyspnea.

PTTM is a disease process in which tumor cells embolize to the pulmonary vasculature, leading to rapid progressive PH in a setting of malignancy [11]. Tumor cell embolization of the pulmonary arterioles, activation of the coagulation cascade, formation of thrombus, and fibrocellular intimal proliferation lead to arteriolar obstruction and, subsequently, to PH. PTTM was first reported by Von Herbay et al. in 1990 as a disease entity, with a reported prevalence of 1–3% [9]. However, the true number of cases is considered higher because patients with PTTM often die before being diagnosed with pathological PTTM. PTTM is challenging to diagnose because laboratory test and chest radiography findings can be unclear. Blood samples show an increase in the levels of D-dimer or fibrinogen breakdown products and cardiac enzymes as right-sided heart failure proceeds [1,12]; however, these markers have low specificity. PTTM can be identified using cytological examination of the aspirate from wedge-shaped pulmonary artery catheters, with a sensitivity and specificity of approximately 90% [4,13,14]. CT-guided lung biopsy, bronchoscopy, or video-assisted thoracoscopic surgery can also be used to confirm PTTM; however, these methods are impractical because they cannot be performed for patients in emergency situations with unstable or severe respiratory distress [15]. Therefore, a high level of suspicion and the use of echocardiography is likely to ensure timely identification of PTTM, which is essential for appropriate management. 

Treatments being tried for PTTM includes systemic chemotherapy, anticoagulant therapy, steroid therapy and treatments for PH [1,16,17,18]. Anti-tumor drugs such as imatinib (a platelet-derived growth factor receptor inhibitor) and bevacizumab (a vascular endothelial growth factor receptor inhibitor) are under trial as they may have a role in reducing vascular remodeling [19]. Two of the nine patients in this study initiated treatment immediately after recognition of PTTM, and they are currently alive. In particular, bedside echocardiography findings of one patient with gastric cancer who required 15 L/min oxygen via a facial mask due to severe dyspnea showed significant RV dilatation with a marked decrease in RV systolic function (Case 6, Appendix C). This patient was hospitalized with a clinical diagnosis of PTTM after chest CT findings confirmed the absence of detectable PTE. She immediately initiated new chemotherapy and anticoagulation treatment. Follow-up echocardiography on admission showed improvement in PH with a slightly decreased RVSP, ranging from 64 mmHg to 52 mmHg, compared to the findings of her bedside echocardiography performed in the ED. Her dyspnea gradually improved, and she was discharged on hospitalization day 17. After approximately 2 months, her NT-proBNP level improved to 107 pg/mL from 5311 pg/mL, and she is currently undergoing chemotherapy at the outpatient clinic. For this patient, PTTM was quickly suspected after echocardiography, leading to prompt treatment.

Although chest CT and radiography are limited in terms of not being able to directly determine heart function, this study summarized the characteristics of the radiological findings in patients with PTTM. Plain chest radiography findings indicated no lung parenchymal lesions that could cause severe respiratory distress among any of the patients; however, cardiomegaly and pulmonary trunk enlargement were observed in some patients. On chest CT, the RV: LV ratio quantified RV dilation, and all patients had a RV: LV ratio of >1, confirming RV dilatation regardless of whether the major pulmonary artery was dilated [20]. In addition, centrilobular GGNs ≤ 3 mm (i.e., small nodules in the center of the secondary pulmonary lobules) were observed in most patients (n = 8). Because these opacities reflect abnormalities in peripheral pulmonary arterial vessels, they are commonly known as an early sign of PTTM and often present as a vascular tree-in-bud sign (n = 5) [21,22]. Peripheral wedge-shaped GGOs were identified in five patients despite these being nonspecific signs of interstitial and airspace edema or interstitial inflammation. Above all, a chest CT scan is particularly important for a clinical diagnosis of PTTM because it confirms the presence or absence of PTE.

In this study, McConnell’s sign was observed in five patients. McConnell’s sign describes a regional pattern of acute RV dysfunction on echocardiography, which was first observed in a cohort of patients with acute PTE [23]. This sign refers to the coexistence of akinesia of the mid-to basal free RV wall with maintained apical contractility, as opposed to the global wall motion abnormalities seen in chronic RV failure [24,25]. Although it is unclear why McConnell’s sign is observed in patients with PTTM, it may be related to the phenomenon of PH associated with microscopic tumor embolism, similar to that seen in PTE. In this study, two patients with suspected PTE after demonstrating acute RV failure were found to have McConnell’s sign on echocardiography. However, after confirming the absence of visible PTE on chest CT, malignancy was suspected to be inversely related to the possibility of PTTM. The condition of these patients deteriorated rapidly and histological confirmation was not possible; however, PTTM was strongly suspected to be related to malignancy (ovarian cancer and metastasis of unknown origin). Therefore, when McConnell’s sign is newly observed in patients with cancer, our findings indicate that PTTM should be considered a differential diagnosis in the absence of PTE.

RV dysfunction may occur in various diseases, such as acute respiratory distress syndrome, RV myocardial infarction, or previously undiagnosed pulmonary artery hypertension [26,27]. Some acute RV failure etiologies can be explained by signs, symptoms, and laboratory tests; however, these findings lack sensitivity and specificity and can be due to several pathologies that produce organ hypoperfusion [28]. There are no precise biochemical markers for acute RV failure [29]; therefore, the diagnostic workup is heavily reliant on a clinical diagnosis supported with imaging. Echocardiography is a valuable tool in the diagnosis of acute RV failure because it can be used to assess RV preload, contractility, and afterload without invasive procedures as well as detect the existence of left-sided cardiac dysfunction. In particular, it is recommended for use as the first diagnostic tool when a patient is clinically unstable or when CT cannot be used. Although an early and definite diagnosis of PTTM is currently challenging, an optimal diagnostic method using echocardiography and a therapeutic strategy is required.

Many clinicians may not consider PTTM as a differential diagnosis as it is a relatively newly defined and rare disease. Based on our study findings concerning patients with PTTM diagnosed for the first time in the ED, we recommend that clinicians should suspect and assess PTTM in their differential diagnosis of patients with acute respiratory distress and right-sided heart failure without PTE, especially in those with a history of malignancy. Delayed diagnosis and treatment of the underlying cause as well as an inability to prevent further right-sided heart failure are all associated with poorer outcomes in patients with PTTM.

### Limitations

This study has several limitations. First, this was a retrospective study involving a small number of patients. As there was no official diagnosis of PTTM in our institution, we could only search for patients with newly diagnosed PH or thrombotic microangiopathy in the ED. Therefore, data on not all patients with PTTM who visited the ED during the study period could be reviewed, and it is difficult to generalize the results of our findings due to the limited patient numbers. Second, all patients were clinically diagnosed based on radiological and echocardiographic findings without cytological confirmation via autopsy. However, all patients experienced a sudden exacerbation of dyspnea within a few days and had no prior history of heart or respiratory issues, even if they had a diagnosis of cancer. Despite the limitations of pathological confirmation, this clinical diagnosis can be considered the most practical and realistic given the clinical picture of patients with PTTM in rapid decline. Finally, in this study, emergency physicians performed echocardiography at the bedside in a short period of time. Patients had severe respiratory symptoms, and it was challenging to obtain optimal images because changing patients’ positions was difficult. Therefore, no comprehensive echocardiographic measurements of the patients could be taken, particularly of other RV parameters such as pulmonary acceleration time and presence of premature systolic notching in the pulse wave Doppler signal of the pulmonary valve.

## 5. Conclusions

PTTM is a rapidly progressing and fatal disease; therefore, it is critical to quickly differentiate it when a patient with cancer and acute respiratory distress presents to the ED. This case series, involving patients first diagnosed with PTTM in the ED through findings of RV failure with PH on bedside echocardiography, may help clinicians establish an earlier diagnosis of PTTM and lead to good patient outcomes through rapid initiation of treatment.

## Figures and Tables

**Figure 1 diagnostics-12-00259-g001:**
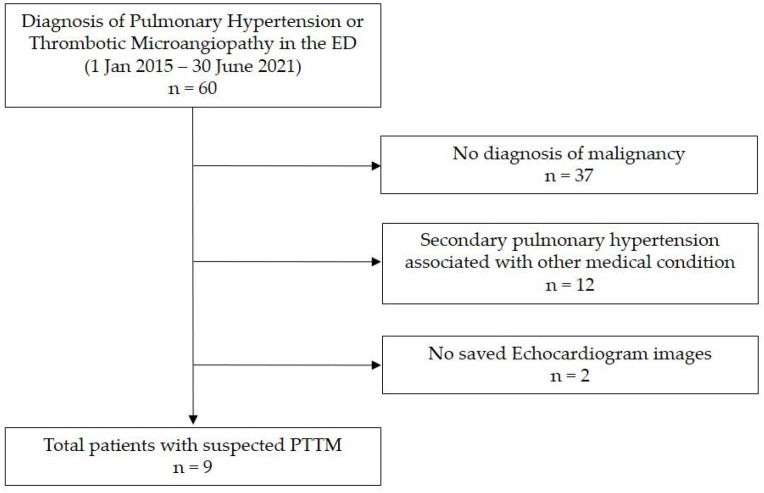
Study patients. Abbreviations: ED, emergency department; PTTM, pulmonary tumor thrombotic microangiopathy.

**Figure 2 diagnostics-12-00259-g002:**
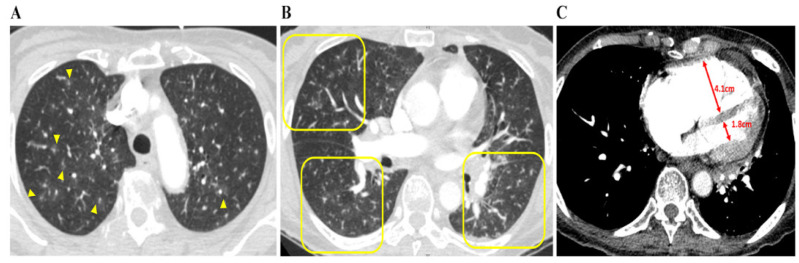
Chest CT findings concerning patients with PTTM. A 73-year-old woman with advanced gastric cancer (Case 8). (**A**) Bilateral lung CT scan results showing disseminated centrilobular GGNs (yellow arrows), (**B**) in which occasional vascular trees-in-bud signs (yellow squares) are noted. (**C**) Right and left ventricle (RV and LV) diameters are 4.1 cm and 1.8 cm, respectively, and the RV: LV ratio is 2:3, indicating that RV dysfunction, despite pulmonary arterial thromboembolism is not being depicted. Abbreviations: CT, computed tomography; GGNs, ground-glass nodules; LV, left ventricle; PTTM, pulmonary tumor thrombotic microangiopathy; RV, right ventricle.

**Figure 3 diagnostics-12-00259-g003:**
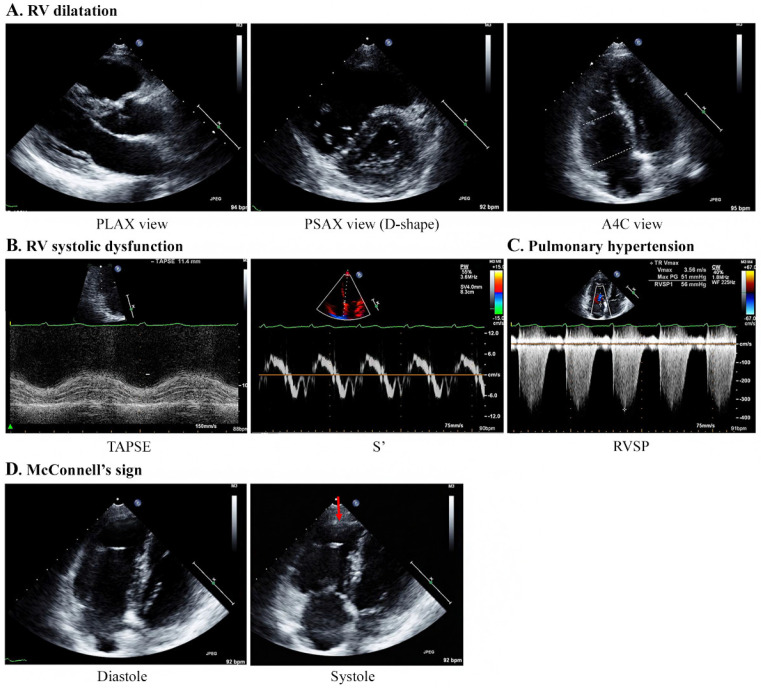
Specific echocardiography features concerning patients with PTTM. A 40-year-old woman with breast cancer (Case 1). (**A**) Echocardiography results indicate RV dilatation with a D-shaped LV (mid/basal RV size: 29/40 mm). (**B**) TAPSE and S’ are 11.4 mm and 7.4 cm/s, respectively, indicating a decrease in RV systolic function. (**C**) The TR Vmax is 3.56 m/s, indicating PH. (**D**) McConnell’s sign, which is an akinesia of the RV mid-free wall except for the apex (red arrow), is observed. Abbreviations: A4C, apical 4 chamber; PLAX, parasternal long axis; PSAX, parasternal short-axis view; PTTM, pulmonary tumor thrombotic microangiopathy; RV, right ventricle; RVSP, right ventricular systolic pressure; S’, lateral tricuspid annulus peak systolic velocity on tissue Doppler imaging; TAPSE, tricuspid annular plane systolic excursion; TR V max, tricuspid regurgitation maximum velocity.

**Table 1 diagnostics-12-00259-t001:** Demographic characteristics and clinical findings concerning patients with PTTM in the ED.

Case	Age (year)	Sex	PH History	Malignancy Diagnosis before ED	Dyspnea Exacerbation (days)	Oxygen Needs (/min)	D-Dimer (μg/mL)	NT-Pro BNP (pg/mL)	Primary Malignancy	ChemoTx. Initiation Time from ED	Death	Time to Death (days)
1	40	F	−	+	3	FM 5 L	3.41	3664	Breast Ca.	−	+	2
2	47	F	−	−	10	FM 15 L	4.28	8153	Ovarian Ca.	−	+	1
3	59	F	−	+	7	HFNC	25.09	4253	Breast Ca.	−	+	3
4	41	F	−	−	3	NC 4 L	60.00	4273	MUO	−	+	3
5	59	F	−	+	1	NC 3 L	6.64	9130	Breast Ca.	24 h	−	−
6	43	F	−	+	2	FM 15 L	16.52	5311	Gastric Ca.	48 h	−	−
7	54	F	−	+	1	NC 5 L	6.18	1728	Breast Ca.	−	+	N/A
8	73	F	−	+	2	FM 8 L	28.11	21,925	Gastric Ca.	−	+	1
9	53	M	−	+	14	FM 5 L	17.98	1520	Bladder Ca.	−	+	3

Abbreviations: Ca, cancer; ChemoTx, chemotherapy; ED, emergency department; F, female; M, Male; FM, facial mask; h, hours; HFNC, high flow nasal cannula; N/A, not available; NC, nasal cannula; MUO, malignancy of unknown origin; PH, pulmonary hypertension; PTTM, pulmonary tumor thrombotic microangiopathy; −, negative or none; +, positive or available.

**Table 2 diagnostics-12-00259-t002:** Radiologic features concerning patients with PTTM.

Case	Plain Radiography			Computed Tomography								
	Cardiomegaly	Pulmonary Trunk Enlargement	Abnormal Findings	RV Inner Cavity (cm)	LV Inner Cavity (cm)	RV: LV Ratio	PTE	Vascular Tree-in-Bud Sign	Centri- Lobular GGNs <3 mm	Peripheral Wedge GGOs	Interstitial Thickening	Consolidation
1	−	+	−	4.1	3.4	1.2	−	−	Diffuse	−	−	−
2	+	+	−	4.3	2.8	1.5	−	−	-	−	−	−
3	−	−	−	3.5	1.9	1.8	−	−	Several random	+	−	−
4	+	−	−	3.2	2.0	1.6	+	+	Diffuse	−	+	−
5	+	−	−	3.3	2.7	1.2	−	+	Diffuse	+	−	−
6	+	+	GGOs, pleural effusion	4.4	3.0	1.5	−	+	Several segmental	+	−	+
7	−	+	−	3.6	3.1	1.2	−	−	Diffuse	+	−	−
8	−	+	Interstitial thickening	4.1	1.8	2.3	−	+	Diffuse	+	+	−
9	−	−	−	3.7	1.6	2.3	−	+	Diffuse	−	−	−

Abbreviations: GGNs, ground glass nodules; GGOs, ground-glass opacification; LV, left ventricle; PTE, pulmonary thromboembolism; RV, right ventricle; −, negative or none; +, positive or available.

**Table 3 diagnostics-12-00259-t003:** Echocardiographic findings concerning patients with PTTM.

Case	LV			RV							
	Size	EF * (%)	Diastolic Function (e/e′)	RV Dilatation ^†^	Size (Base/Mid) (mm)	D-Shape	TAPSE (mm)	S′ (cm/s)	TR Vmax (m/s)	RVSP (mmHg)	McConnell’s Sign
1	Normal	62	7.1	Moderate	40/29	+	11.4	7.7	3.56	56	+
2	Normal	Normal	N/A	Severe	N/A	+	8	N/A	3.6	60	−
3	Normal	67	N/A	Severe	42/33	+	21.1	11.9	4.5	87	+
4	Normal	63	N/A	Severe	42/39	+	16.4	N/A	3.7	72	+
5	Normal	57	N/A	Severe	50/34	+	17	11.3	3.82	64	+
6	Normal	58	5.7	Moderate	44/37	+	15.3	N/A	3.43	62	+
7	Normal	56	8.9	Moderate	45/37	+	N/A	N/A	3.49	54	−
8	Normal	Normal	N/A	Severe	N/A	+	N/A	N/A	N/A	N/A	−
9	Normal	56	5.8	Moderate	42/31	+	13.5	10.6	4.91	93	−

* LV systolic function (EF) was evaluated via modified Simpson’s method or visual estimation. ^†^ RV dilatation: mild, RV is less than 2/3 of LV; moderate, RV is greater than 2/3 of LV and less than LV; severe, RV is greater than LV. Abbreviations: EF, ejection fraction; IVC, inferior vena cava; LV, left ventricle; N/A, not applicable; RV, right ventricle; RVSP, right ventricular systolic pressure; TAPSE, tricuspid annular plane systolic excursion; TR V max, tricuspid regurgitation maximum velocity, −, negative or none; +, positive or available.

## Data Availability

Data related to this study cannot be sent to the outside due to information security policies in the hospital.

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
