# Peer review of "Echocardiographic Assessment of Patients with Pulmonary Tumor Thrombotic Microangiopathy First Diagnosed in the Emergency Department"

_diagnostics, 2022, doi:10.3390/diagnostics12020259_

Round 1

Reviewer 1 Report

This paper reported echocardiographic data of 9 patients with pulmonary tumor thrombotic microangiopathy which were firstly diagnosed in the emergency. The topic is interesting, this type of disease are rare but fatal, emergency echocardiography were convenient and useful to provide important dynamic cardiovascular information, which is valuable for clinical diagnosis and prognosis judgement. And the whole manuscript was well organized.

Author Response

Comments and Suggestions for Authors

This paper reported echocardiographic data of 9 patients with pulmonary tumor thrombotic microangiopathy which were firstly diagnosed in the emergency. The topic is interesting, this type of disease are rare but fatal, emergency echocardiography were convenient and useful to provide important dynamic cardiovascular information, which is valuable for clinical diagnosis and prognosis judgement. And the whole manuscript was well organized.

Response: We appreciate your review and all comments. Our study is significant in that echocardiography can be used as a fast and useful tool to establish an earlier diagnosis of PTTM in emergency department. It is hoped that this article will contribute to the early diagnosis of PTTM and the initiation of appropriate treatment, thereby positively affecting the patient's prognosis.

Reviewer 2 Report

Although the article is partially original, it seems quite instructive for the readers with the presentation of interesting facts. Written in clear language and supported by echocardiographic images. The manuscript is acceptable for printing in its current form.

Author Response

Comments and Suggestions for Authors

Although the article is partially original, it seems quite instructive for the readers with the presentation of interesting facts. Written in clear language and supported by echocardiographic images. The manuscript is acceptable for printing in its current form.

Response: Thank you for taking the time to leave your thoughts. PTTM is a rare but fatal condition, and echocardiography in the emergency department can help with early identification. In this regard, our study may be helpful by identifying specific PTTM echocardiographic findings.

Reviewer 3 Report

I would like to congratulate the authors for gathering such data and writing this article, which could be a great help for clinicians to diagnose PTTM. However, I have some questions and comments, which I think would enhance the quality of the article.

-In the methods section, it is stated that "there is no accurate diagnostic code for PTTM", which I would presume when patients being are hospitalized they would be hospitalized with other more prevalent diagnoses such as PTE. However, in the appendix, most of the patients are hospitalized with suspected or confirmed diagnosis of PTTM. In case PTTM was not the primary diagnosis, it would be nice to state what was the other primary differential diagnosis.

-It would enrich the article more if you could provide arterial blood gas analyses, if available.

-in the last paragraph of the discussion, it is stated that "...relatively new and rare disease", I would suggest you replace "new" with "newly defined."

-Last but not least, it seems initiation of chemotherapy early in the process of patient admission plays an important role in the prognosis of PTTM. Although we are not to judge why the chemotherapy was not initiated for the rest of the patient, because each patient undergo individually tailored diagnostic and therapeutic procedures, how come the other patients did not undergo chemotherapy when PTTM was suspected?

Author Response

Response to Reviewer 3 Comments

Comments and Suggestions for Authors

I would like to congratulate the authors for gathering such data and writing this article, which could be a great help for clinicians to diagnose PTTM. However, I have some questions and comments, which I think would enhance the quality of the article.

Thank you for your feedback and suggestions. It is hoped that this article will contribute to the early diagnosis of PTTM and the initiation of appropriate chemotherapy, thereby positively affecting the patient's prognosis.

Point 1: In the methods section, it is stated that "there is no accurate diagnostic code for PTTM", which I would presume when patients being are hospitalized they would be hospitalized with other more prevalent diagnoses such as PTE. However, in the appendix, most of the patients are hospitalized with suspected or confirmed diagnosis of PTTM. In case PTTM was not the primary diagnosis, it would be nice to state what was the other primary differential diagnosis.

Response 1: There was no officially designated diagnostic code for PTTM in our hospital's Electronic Medical Record (EMR). Therefore, patients were usually hospitalized under their primary malignancy diagnostic code even though PTTM was highly suspected, and the diagnostic code for "pulmonary hypertension" or "thrombotic microangiopathy" were used instead of PTTM in some cases. That is why we have screened patients with pulmonary hypertension and thrombotic microangiopathy codes on EMR when we began this study to collect PTTM cases.

Point 2: It would enrich the article more if you could provide arterial blood gas analyses, if available.

Response 2: Thanks for your recommendation. We considered presenting the ABGA data in a table, but most of the patients had severe respiratory problems, and oxygen was applied during blood sampling for ABGA, which could make the ABGA results difficult to interpret. The table below shows the ABGA results of 9 patients. Only 2 patients underwent ABGA in room air. Therefore, we made an extra table of arterial blood gas analyses and attached it as part of appendix.

Appendix A. The results of arterial blood gas analyses concerning patients with PTTM.

Case

Hb

(g/dL)

pH

pCO2

(mmHg)

pO2

(mmHg)

HCO3-

(mmol/l)

BE

(mmol/l)

SO2
(%)

Oxygen applied during ABGA sampling (/min)

1

10.5

7.539

26.1

83.7

21.8

0.1

95.7

FM 5 L

2

13.8

7.464

16.6

58.7

11.6

-9.2

89.5

RA

3

10.7

7.482

27.4

85.4

20.0

-2.4

95.3

NC 2L

4

10.0

7.48

24

97

17.9

-3.8

98.6

NC 3 L

5

13.2

7.42

30

102

19.5

-3.9

98.3

NC 3 L

6

10.5

7.47

28

35

20.4

-2.4

62.5

RA

7

8.3

7.535

24.0

117.3

19.8

-2.1

97.9

NC 5 L

8

11.6

7.49

22

86

16.8

-4.5

97.9

FM 8 L

9

13.2

7.482

27.9

116.8

20.4

-1.8

97.8

FM 5 L

Hb, hemoglobin; pH, acidity; pCO2, partial pressure of carbon dioxide; pO2, partial pressure of oxygen; BE, base excess; SO2, oxygen saturation; ABGA, arterial blood gas analyses; FM, facial mask; RA, room air; NC, nasal cannula

Point 3: in the last paragraph of the discussion, it is stated that "...relatively new and rare disease", I would suggest you replace "new" with "newly defined."

Response 3: Thanks for your suggestion. We revised the sentence by replacing “new” with “newly defined”.

Point 4: Last but not least, it seems initiation of chemotherapy early in the process of patient admission plays an important role in the prognosis of PTTM. Although we are not to judge why the chemotherapy was not initiated for the rest of the patient, because each patient undergo individually tailored diagnostic and therapeutic procedures, how come the other patients did not undergo chemotherapy when PTTM was suspected?

Response 4: It's true that getting started on chemotherapy as soon as possible is crucial. However, with the exception of two patients in our assessment, most patients' conditions deteriorated so quickly that proper treatment could not be started in time. One patient died in the ED while awaiting hospitalization due to sudden cardiac arrest, and the other patient died in the hospital while preparing chemotherapy. Multi-organ failure was also so severe in certain cases that chemotherapy was not an option. Another patient was diagnosed with PTTM in our emergency room and transferred to the hospital where she was receiving treatment, where she died a few hours after arriving. A breast cancer patient with brain metastasis was rushed out of the hospital because she refused to be treated any longer. Only two patients were able to endure chemotherapy, resulting in reduced oxygen demand and favorable results. PTTM is a rapidly progressing and fatal disease. So, we hope that early recognition of RV failure with PH on bedside echocardiography will enable clinicians to establish an earlier diagnosis of PTTM and lead to good patient outcomes through rapid initiation of treatment.
